# Carrier Rate and Mutant Allele Frequency of GM1 Gangliosidosis in Miniature Shiba Inus (Mame Shiba): Population Screening of Breeding Dogs in Japan

**DOI:** 10.3390/ani12101242

**Published:** 2022-05-12

**Authors:** Shahnaj Pervin, Md Shafiqul Islam, Yamato Yorisada, Aya Sakai, Shimma Masamune, Akira Yabuki, Tofazzal Md Rakib, Shinichiro Maki, Martia Rani Tacharina, Osamu Yamato

**Affiliations:** 1Laboratory of Clinical Pathology, Joint Faculty of Veterinary Medicine, Kagoshima 890-0065, Japan; s.pervin30@yahoo.com (S.P.); si.mamun@ymail.com (M.S.I.); yabu@vet.kagoshima-u.ac.jp (A.Y.); rakibtofazzal367@gmail.com (T.M.R.); k6993382@kadai.jp (S.M.); martia.rt@fkh.unair.ac.id (M.R.T.); 2Department of Pathology and Parasitology, Faculty of Veterinary Medicine, Chattogram Veterinary and Animal Sciences University, Khulshi, Chattogram 4225, Bangladesh; 3P’s-First Co. Ltd., 1-24-12 Meguro, Meguro-ku, Tokyo 153-0063, Japan; yorisada@pfirst.jp (Y.Y.); sakai@pfirst.jp (A.S.); masamune@pfirst.jp (S.M.); 4Faculty of Veterinary Medicine, Airlangga University, Mulyorejo, Surabaya 60115, Indonesia

**Keywords:** carrier rate, mutant allele frequency, Mame Shiba Inu, GM1 gangliosidosis, canine *GLB1* gene, dog breeding

## Abstract

**Simple Summary:**

GM1 gangliosidosis, a progressive neurodegenerative lysosomal storage disease, occurs in the Shiba Inu breed due to the *GLB1*:c.1649delC (p.P550Rfs*50) mutation. Previous surveys performed of the Shiba Inu population in Japan showed that the carrier rate was 1.02–2.94%. Currently, a miniature type of the Shiba Inu called “Mame Shiba”, bred via artificial selection to yield smaller individuals, is becoming more popular than the standard type. A GM1 gangliosidosis mutation survey has yet to be carried out in the Mame Shiba population. Therefore, this study aimed to determine the frequency of the mutant allele and carrier rate of GM1 gangliosidosis in the Mame Shiba breed. We surveyed 1832 adult Mame Shiba Inus used for breeding across 143 Japanese kennels using a genotyping assay. As a result, nine Mame Shiba Inus were found to be carriers, indicating that the carrier rate was 0.49% (corresponding mutant allele frequency = 0.00246). This study demonstrates that there are certain Mame Shiba Inus carrying the mutation for GM1 gangliosidosis and that their breeding should be prevented or controlled.

**Abstract:**

GM1 gangliosidosis is a progressive, recessive, autosomal, neurodegenerative, lysosomal storage disorder that affects the brain and multiple systemic organs due to an acid β-galactosidase deficiency encoded by the *GLB1* gene. This disease occurs in the Shiba Inu breed, which is one of the most popular traditional breeds in Japan, due to the *GLB1*:c.1649delC (p.P550Rfs*50) mutation. Previous surveys performed of the Shiba Inu population in Japan found a carrier rate of 1.02–2.94%. Currently, a miniature type of the Shiba Inu called “Mame Shiba”, bred via artificial selection to yield smaller individuals, is becoming more popular than the standard Shiba Inu and it is now one of the most popular breeds in Japan and China. The GM1 gangliosidosis mutation has yet to be surveyed in the Mame Shiba population. This study aimed to determine the frequency of the mutant allele and carrier rate of GM1 gangliosidosis in the Mame Shiba breed. Blood samples were collected from 1832 clinically healthy adult Mame Shiba Inus used for breeding across 143 Japanese kennels. The genotyping was performed using a real-time PCR assay. The survey found nine carriers among the Mame Shibas, indicating that the carrier rate and mutant allele frequency were 0.49% and 0.00246, respectively. This study demonstrated that the mutant allele has already been inherited by the Mame Shiba population. There is a risk of GM1 gangliosidosis occurrence in the Mame Shiba breed if breeders use carriers for mating. Further genotyping surveys are necessary for breeding Mame Shibas to prevent the inheritance of this disease.

## 1. Introduction

GM1 gangliosidosis is a progressive, recessive, autosomal, neurodegenerative, lysosomal storage disorder that affects the brain and multiple systemic organs due to an acid β-galactosidase (GLB1, EC 3.2.1.23) deficiency encoded by the *GLB1* gene [1,2]. The disease is characterized by an abnormal accumulation of β-linked galactose-containing glycoconjugates in the central nervous system, including the glycosphingolipid GM1 ganglioside, resulting in the premature death of affected individuals due to brain damage from progressive neurological malfunctions.

GM1 gangliosidosis occurs in humans (Online Mendelian Inheritance in Man (OMIM) 230500, 230600, and 230650) and several other animal species (Online Mendelian Inheritance in Animals (OMIA) 000402), including dogs, cats, cattle, sheep, emus, American black bears, and murine models [1]. Naturally occurring GM1 gangliosidosis has been reported in many dog breeds, including mixed Beagles [3], English Springer Spaniels [4], Portuguese Water Dogs [5], Alaskan Huskies [6], a mixed breed dog [7], and Shiba Inus [8]. To date, three different mutations have been identified in the canine *GLB1* gene that can cause the disease in Portuguese Water Dogs [9], Shiba Inus [10], and in Alaskan Huskies [11]. According to the OMIA website (https://www.omia.org/ (accessed on 23 April 2022)), the causative mutations are located in the canine chromosome 23 (CanFam3.1), specifically, g.3754313G>A (c.179G>A, p.R60H) in Portuguese Water Dogs, g.3796317delC (c.1649delC, p.P550Rfs*50) in Shiba Inus, and g.3796356_3796374dup (c.1688_1706dup, p.T570Pfs*22) in Alaskan Huskies.

The Shiba Inu breed, also called the standard Shiba (Table 1 and Figure 1), is an ancient and native basal spitz breed in Japan [12] and it was designated as a protected species in 1936 [13]. The breed is genetically similar to that of wolves and differs from European dog breeds [14]. The breed is indigenous to Japan and has become one of the most popular breeds in the country, with thirty to forty thousand puppies registered every year in Japan alone [15]. Shiba Inus have been transported worldwide and are bred and maintained as a standard breed in many countries. However, in recent years, artificial selection has been used while breeding standard Shiba Inus to produce puppies with a smaller body size, which are called “Mame Shiba”, or the miniature Shiba (Table 1 and Figure 1) [16]. “Mame” is a Japanese word that means “bean”, representing “small”. The Mame Shiba is bred from the standard Shiba Inu in order to maintain their purebred status, and both breeds share some common traits, such as coat color. Whole genome sequencing indicates that there may be a link between specific candidate genes and body size [16]. Although the Shiba Inu and other traditional Japanese breeds, such as the Akita, Kishu, Shikoku, Kai, and Hokkaido (Table 1), are approved by the Japan Kennel Club (JKC), a kennel club certified by the Federation Cynologique Internationale (FCI), and the Nihon-ken Hozonkai (NIPPO), a kennel club especially for traditional Japanese dog breeds, the Mame Shiba has not been approved by any of these organizations. The Mame Shiba is currently approved and standardized separately by the Kennel Club of Japan (KCJ) and the Nihon Mame Shibaken Association (NMSA), each of which issue specific pedigree papers to Mame Shiba Inus. The Mame Shiba is becoming more popular than the standard Shiba Inu and is now one of the most popular breeds in Japan and China [16], but the data about the number of registered Mame Shibas are not publicly available.

In 2008, a small-scale molecular survey of GM1 gangliosidosis in 68 standard Shiba Inus was carried out in northern Japan, which found two carrier dogs, indicating a carrier rate of 2.94% [15]. Following this, a large-scale molecular epidemiological survey was carried out of 590 standard Shiba Inus across all districts of Japan in 2013, which found six carriers, indicating an average carrier rate of 1.02% in Japan and 2.27% in the Kinki district [13]. There is a high probability of transferring the mutant allele to the Mame Shiba as it is selected and bred from the standard Shiba Inu. However, the GM1 gangliosidosis mutation has yet to be surveyed in the Mame Shiba population. Therefore, it is necessary to survey the Mame Shiba population to learn the current carrier rate and mutant allele frequency and plan an effective strategy for breeders to control GM1 gangliosidosis in this new, popular canine breed. In this study, we performed a large-scale molecular survey of GM1 gangliosidosis in the Mame Shiba population in Japan.

## 2. Materials and Methods

The experiments conducted in this study were performed in accordance with the guidelines regulating animal use and ethics at Kagoshima University (no. VM15041; approval date: 29 September 2015) and oral informed consent was obtained from the participating breeders.

### 2.1. Sample Collection and Genotyping

From February 2019 to April 2022, whole blood samples (≤0.3 mL) were randomly collected from 1832 clinically healthy adult Mame Shiba Inus (439 males and 1393 females), across 143 kennels in the Kyushu to Kanto districts, Japan, that were born between 2012 and 2021 and are used for breeding. The blood samples were spotted onto Flinders Technology Associates filter papers (FTA card; Whatman International Ltd., Piscataway, NJ, USA) and stored in a refrigerator at 4 °C ready for DNA extraction. DNA was extracted from discs punched out of these FTA cards following appropriate treatment, as previously described [17]. The genotypes of the dogs were determined using real-time PCR, as previously reported [17].

### 2.2. Statistical Analysis

The allele frequencies obtained in this study were analyzed using a Chi-square test for Hardy–Weinberg equilibrium. The deviations between the measured and expected values were regarded as statistically significant at *p* < 0.05. Differences in the carrier rates between the Mame Shiba population in this study and the standard Shiba Inu populations in previous studies [13,15] were statistically analyzed using Fisher’s exact test, with *p* < 0.05 considered to be a statistically significant difference. Statistical analyses were performed using R software.

## 3. Results

The real-time PCR clearly determined the genotypes of all the samples. Genotyping of Mame Shiba Inus revealed that among the 1832 dogs surveyed, there were 1823 homozygous wild-type dogs, nine heterozygous carriers, and no homozygous mutant dogs. Based on these observations, we estimated a carrier rate of 0.49%. The corresponding mutant allele frequency is 0.00246, indicating expected frequencies of homozygous wild-type, heterozygous carrier, and homozygous mutant genotypes of 0.995, 0.00490, and 0.00000603, respectively. A Chi-square test analysis (χ^2^ = 0.0111; df = 2; *p* value = 0.995) indicates that these three genotypes were in Hardy–Weinberg equilibrium.

Differences in the carrier rates between the populations of the Mame Shiba in this study (0.49%, 9/1832) and the standard Shiba Inu in our previous studies (2.94%, 2/68 [15]; 1.02%, 6/590 [13]) were analyzed statistically using Fisher’s exact test. There were no significant differences between the carrier rates in the Mame Shiba in this study and the standard Shiba Inu in either study in 2008 (*p* = 0.222) or 2013 (*p* = 0.0564).

## 4. Discussion

Dogs have been associated with humans longer than any other domestic animals and they play a number of important roles in modern society [18]. Many researchers are working to localize genes of interest in dogs, particularly disease-related genes, to understand inherited diseases in humans and to guide dog breeding programs in dogs to improve their health and welfare [19]. Therefore, knowledge of the inheritance risk for different diseases in a typical dog breed and identifying which diseases spread between dog breeds is valuable for both veterinary care and for breeding healthy dogs; this includes GM1 gangliosidosis in the Shiba Inu and the Mame Shiba.

The clinical signs of GM1 gangliosidosis in the Shiba Inu are very severe, with the onset of neurological signs at 5 to 6 months of age, which worsen gradually until death due to the accumulation of GM1 ganglioside in the central nervous system [8,20,21]. The age at which euthanasia is requested by the owners or the occurrence of sudden natural death is from 12 to 15 months of age [8], or by 18 months old at the latest. Long-term care for affected dogs with such devastating neurological signs may be a considerable mental and physical burden to their owners. Therefore, even a small number of dogs with GM1 gangliosidosis could have a major impact on owners, breeders, the various companies involved in selling dogs, and kennel clubs. Therefore, the appearance of affected dogs and a high frequency of the GM1 gangliosidosis mutant allele are undesirable in the Mame Shiba breed.

This study revealed that the carrier rate and mutant allele frequency of GM1 gangliosidosis in the breeding population of the Mame Shiba were 0.49% and 0.00246, respectively. They are relatively low compared to the figures from 2008 (2.94% and 0.0147) and 2013 (1.04% and 0.00508) in the standard Shibas [13,15], but the difference is not significant. This suggests that the Mame Shiba has already inherited GM1 gangliosidosis through the establishment of the breed from the standard Shiba Inu. Given the lethality of this disease and the popularity of the Mame Shiba, the corresponding mutant allele frequency (0.00246) is deemed sufficiently high to warrant measures for disease control and prevention, as in the standard Shiba Inu.

In Japan, molecular epidemiological surveys have previously been performed for several canine inherited diseases to determine the associated carrier rates and mutant allele frequencies and thereby evaluate the necessity of prevention measures [22,23,24]. Among these diseases, lethal disorders characterized by progressive neurological disfunctions include Sandhoff disease in Toy Poodles (carrier rate = 0.20%) [22], neuronal ceroid lipofuscinosis (NCL) in Border Collies (8.11%) [23], and NCL in Chihuahuas (1.29%) [24]. Similar to Shiba Inus, Toy Poodles and Chihuahuas are also particularly popular canine breeds in Japan; according to the JKC, from 2008 to 2021, they were the first and second most commonly registered breeds in Japan, respectively. Consequently, underlying lethal diseases in these popular breeds could be expected to have wide-ranging implications, such as those associated with GM1 gangliosidosis in standard Shiba Inus and Mame Shibas. Indeed, standard Shiba Inus affected by GM1 gangliosidosis have continued to be detected at the rate of 1–5 dogs per year [13]. In addition, GM1 gangliosidosis has been diagnosed in two standard Shiba Inus in China and Korea, respectively (personal information, O.Y.). The Mame Shiba is becoming more popular in Japan and China [16] and, therefore, the prevention and eradication of GM1 gangliosidosis in Mame Shibas is recommended by preventing the reproduction of affected dogs and reducing the number of carriers. A comprehensive screening of all puppies, for example, blanket screening, is considered ineffective and prohibitively expensive in a large population such as Chihuahuas [24]. Continued genotyping should be performed on breeding Mame Shibas, and breeders should appropriately manage mating. This prevention strategy would be more effective and lower cost, especially in the large populations of Shiba Inus and Mame Shibas.

Besides GM1 gangliosidosis, a causative mutation for Sandhoff disease has been identified in standard Shiba Inus in the United States [25,26]. The Shiba Inu is genetically predisposed to glaucoma [27], which is potentially associated with certain genes [28]. The Shiba Inu is also predisposed to chronic enteropathy [29,30] and atopic dermatitis, which are also related to certain genetic backgrounds [30,31]. In addition, the Shiba Inu has a special phenotype that usually has no clinical sign, but they are more susceptible to onion-induced hemolytic anemia and one of its causative agents (NPTS) than others at the same dosage [32]. It is important for the total health management of the breed to clarify whether these genetic characteristics have been transferred to the Mame Shiba. However, body size, inbreeding, and deleterious morphologies should also be considered [33]. It is important not only to monitor inherited or genetic diseases, but also the disorders caused by the potentially excessive miniaturization of the Mame Shiba.

## 5. Conclusions

The results of this study show that the carrier rate of GM1 gangliosidosis in the Mame Shiba in Japan is currently 0.49%, and, given the lethality of this disease and the popularity of this breed, the corresponding mutant allele frequency (0.00246) is deemed sufficiently high to warrant measures for disease control and prevention. Ideally, continued genotype surveying should be performed on breeding Mame Shibas reared by breeders who undertake appropriate mating management.

## Figures and Tables

**Figure 1 animals-12-01242-f001:**
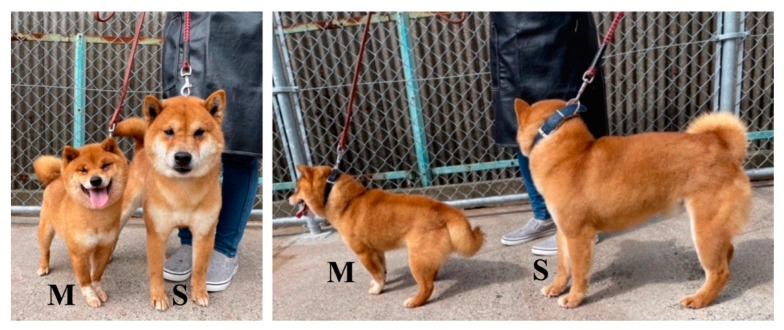
Typical appearance of a Shiba Inu (S: standard type) and a Mame Shiba Inu (M: miniature type). The coat color of these dogs is red, the most popular among the four coats found in these breeds (red, black-and-tan, sesame, and white).

**Table 1 animals-12-01242-t001:** Withers height of Japanese dog breeds.

Body Size	Breed *	Club **	Sex	Withers Height (cm)		
				Standard/Ideal	Range	Limitation
Large	Akita	NIPPO	Male	67	64–70	ND
			Female	61	58–64	
Middle	Kai, Hokkaido	NIPPO	Male	52	47–55	ND
			Female	49	44–52	
	Kishu, Shikoku	NIPPO	Male	52	49–55	ND
			Female	49	46–52	
Small	Shiba	NIPPO	Male	39.5	38–41	ND
			Female	36.5	35–38	
Miniature	Mame Shiba	KCJ	Male	ND	30–34	≥25
			Female	ND	28–32	
		NMSA	Male	30	25–34	Caution (<25)
			Female	28	25–32	

* The Japanese traditional breeds Akita, Kishi, Shikoku, Kai, Hokkaido, and Shiba are approved by the Nihon-ken Hozonkai (NIPPO: https://www.nihonken-hozonkai.or.jp (accessed on 23 April 2022)) and the Japan Kennel Club (JKC: https://www.jkc.or.jp (accessed on 23 April 2022)), which is certified by the Federation Cynologique Internationale (FCI: http://www.fci.be/ (accessed on 23 April 2022)). These two kennel clubs issue pedigree papers to registered dogs, but the Mame Shiba is not approved by either kennel club. ** The standards of withers height for Japanese breeds (except for Mame Shiba) are provided by NIPPO. The Mame Shiba is approved and standardized separately by the Kennel Club of Japan (KCJ: http://www.kcj.gr.jp/index.html (accessed on 23 April 2022)) and the Nihon Mame Shibaken Association (NMSA: https://nmsa.jpn.com (accessed on 23 April 2022)), which both issue specific pedigree papers to registered Mame Shibas. ND: not determined.

## Data Availability

Not applicable.

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
