# Peer review of "Carrier Rate and Mutant Allele Frequency of GM1 Gangliosidosis in Miniature Shiba Inus (Mame Shiba): Population Screening of Breeding Dogs in Japan"

_animals, 2022, doi:10.3390/ani12101242_

Round 1

Reviewer 1 Report

Carrier rate of GM1 gangliosidosis in ‘mame Shiba’ population in Japan. Methodology was right way and results was correctly obtained. An important topic and interesting data were addressed in this manuscript. The data from this study are worthy of publication in Animal, which is important for all breeders and most of veterinarians working on small animal practices.

Major points

Carrier rate was mainly showed and discussed. But the mutant allele was showed only gene frequency. Why the authors did not show genotype frequency of the homozygous mutant as well as carrier rate.

There is no discussion of the entire population of Mame Shiba dogs in Japan. How many Mame Shiba dogs are in Japan. The gene frequency of the mutant allele seems to be low, but I wonder if there are dogs with this disease in Japan.

The title of this study was ‘carrier Rate of GM1...’. The Authors should explain the importance of the carrier for this disease in introduction.

Minor points

Line 161: delete ‘history’.

Line 164: natural death?

Line 168: ‘can’ may be ‘could’

Line 168: companies of what?

Line 169-170: Please consider to delete ‘a high frequency’. An undesirable thing may be only affected dogs.

Line 178: ‘sufficiently high to warrant measures’ seems subjective. If the authors address that the frequency is high, please provide the reason or evidence why the authors determined that the frequency was high.

Line 196: ‘to preventing the production of affected dogs’: Should discuss entire population of Mame Shiba in Japan. How many mame shiba dogs are estimated to be affected in Japan?

Line 204-207: Difficult to understand the meaning of this sentence. Please rephrase.

Line 207: no clinical manifestation of what?

Line203-214: This paragraph does not indicate any relevance to the results of this study. The authors should explain more about the relevance of the results of this study.

Reviewer 2 Report

Review of 1717854

Carrier Rate of GM1 Gangliosidosis in Miniature Shiba Inus  (Mame Shiba): Population Screening of Breeding Dogs in Japan

General Comments

This is an important well written report.  It clearly demonstrates that the mutant allele responsible for gangliosidosis is present in the newly emerging miniature variety of this dog.  Given the catastrophic end point of this disease the authors also sensibly suggests a screening program amongst breeders and appropriate management of their breeding stock so that the mutation is kept out of the miniature animals.

The validity of all finings in this report hang on the accuracy of the test for the alleles.  Given it is a PCR test I assume it is very accurate.  But there were likely some samples that did not produce a result because the sample did amplify in PCR.  The authors should tell us what that rate was so it’s cost can be factored into screening programs.

My main comments are about the sampling regime and the rate of the mutant alleles in the new population.  The authors rightly talked only about the rate of occurrence in the sample, not the population.  In order to infer that rate we would have to be certain that the sample of dogs was representative of the whole population.  That could be done and it is a relatively standard calculation, but given the low rate of occurrence in the sample, the number that would need to be sampled would be large and probably prohibitively expensive.

I would have liked some more explanation in the report of how the kennels were selected for sampling and how big a sample of all the breeding kennels that was.  The authors could have informed the readers whether they had deliberately sampled (or avoided) Shiba Inu “hotspots”.  I would also have preferred to see some comparison of the rate of the mutant alleles in the Mame Shiba population and the Shiba Inu population on a regional basis.  If the rate in the Mame Shiba was high in one region compared to another and was also high in the Shiba Inu population in that area we could have more confidence that the sample reflected the population accurately. 

I was convinced by this report that a screening program for breeders in this new dog variety is warranted.  Especially before the variety grows large and the rate of mutant alleles spreads internationally out of hand.  But others may not be.  So I would have liked a rudimentary cost/benefit analysis of a screening program.  These authors are uniquely in a position to quantify the cost of collecting and processing samples so those data are probably readily  available.  The benefits are harder to come by and not all monetary as the authors rightly mentioned.  But an approximation could be made.

For example if we assume the carrier rate is the same in the whole population as it was in this sample, then practitioners would have to sample about 200 dogs to find one carrier.  The authors could estimate the lifetime cost managing and treating one case of gangliosidosis by consulting experienced veterinary practitioners and getting their estimate. 

The two costs, even though not exact estimates, could be compared to show the monetary benefit.  For a report of this nature that would be appropriate.  I suspect this cost/benefit would be highly favourable for the screening program given the fatal and prolonged effects of this disease.  All the while not forgetting about the non-monetary benefits such as prevention of pain and suffering for the dogs and owners.

Reviewer 3 Report

Pervin S. et al. discussed the frequency of mutant allele and carrier rate GM1 gangliosidosis in the Mame Shiba breed. GM1 gangliosidosis is a hereditary lysosomal storage disorder caused by abnormal accumulation of GM1 ganglioside, a lipid that progressively destroys nerve cells in the brain and spinal cord. The disease is progressive lethal and occurs in the Shiba Inu breed due to the c.1649delC (p.P550 Rfs*50) mutation in in the b-galactosidase gene (GLB1). GM1 is inherited in an autosomal recessive fashion and dogs with one normal and one affected gene (carriers) are normal and show no sign of the disease. The communication is well written, organized and comprehensive. The study was performed on a large number of subjects.  The genotyping could be useful to quickly identify carriers to avoid breeding these together.

Considering the following papers, I think the novelty of this communication is quite limited:

  1. Yamato, O., Jo, E.O., Chang, H.S., Satoh, H., Shoda, T., Sato, R., Uechi, M., Kawasaki, N., Naito, Y., Yamasaki, M. and Maede, Y., 2008. Molecular screening of canine GM1 gangliosidosis using blood smear specimens after prolonged storage: detection of carriers among Shiba dogs in northern Japan. Journal of veterinary diagnostic investigation, 20(1), pp.68-71.
  2. Chang, H.S., Arai, T., Yabuki, A., Hossain, M.A., Rahman, M.M., Mizukami, K. and Yamato, O., 2010. Rapid and reliable genotyping technique for GM1 gangliosidosis in Shiba dogs by real-time polymerase chain reaction with TaqMan minor groove binder probes. Journal of veterinary diagnostic investigation, 22(2), pp.234-237.

Please, find my minor comments below:

Material and Methods: The experiments should be easily reproducible, please indicate the sequence of specific primer pairs used to perform real time PCR and generally expand this section. The mutation, (I am assuming c.1649delC (p.P550 Rfs*50) was identified by aligning and comparing the sequence data online using a specific program? Please report the software.

Statistical Analysis: I think the statistical software should be reported.
